# Total Esophagogastric and Cologastric Dissociation in Neurologically Normal Children: Systematic Review

**DOI:** 10.3390/children9070999

**Published:** 2022-07-02

**Authors:** Elisa Negri, Riccardo Coletta, Kejd Bici, Adrian Bianchi, Antonino Morabito

**Affiliations:** 1Department of Pediatric Surgery, Meyer Children’s Hospital, University of Florence, 50139 Florence, Italy; riccardo.coletta@meyer.it (R.C.); kejd.bici@meyer.it (K.B.); antonino.morabito@unifi.it (A.M.); 2Department of Neuroscience, Psychology, Drug Research and Child Health (NEUROFARBA), University of Florence, 50121 Florence, Italy; 3School of Heath and Society, University of Salford, Manchester M6 6PU, UK; 4Royal Manchester Children’s Hospital, Manchester University NHS Foundation Trust, Manchester M13 9WL, UK; bianchi54@gmail.com

**Keywords:** gastroesophageal reflux, total esophagogastric dissociation, esophageal atresia, neurologically normal children

## Abstract

Total esophagogastric dissociation (TEGD) was first described by Bianchi as a definitive procedure for gastroesophageal reflux disease (GERD) in neurologically impaired children. In the last 20 years, different centers extended the indication to neurologically normal (NN) patients with GERD associated with congenital or acquired esophageal anomalies. The aim of this paper is to analyze the role of TEGD in this cluster of patients. A PubMed and Google Scholar search was conducted. All cases of NN children who underwent TEGD for GERD were collected. Patient characteristics and outcomes were analyzed. Complications were classified according to Clavien–Dindo classification. Forty-eight children were identified. In 56.25%, TEGD was the first anti-reflux procedure, while in 43.75% it was performed after failed fundoplications. Mean follow-up was 5.5 years. Mortality related to surgery was 2.08%. All of the survivors improved their condition, with resolution of GERD and weight gain. In addition, 50% of children weaned off enteral nutrition, with 14.6% having their gastrostomy removed, while 41.67% maintained partial enteral supplementation. Respiratory symptoms almost disappeared in 54.17% of patients. This review suggests that TEGD can also be considered for NN children where conventional methods seem insufficient to control reflux and preserve pulmonary function. Nevertheless, long-term follow-up is still required.

## 1. Introduction

Gastroesophageal reflux disease (GERD) is a common condition, defined by the passage of gastric contents into the esophagus causing troublesome symptoms and/or complications, affecting up to 3.3% of the pediatric population [1]. Its incidence is widely variable, reaching >77% in level IV–V neurologically impaired (NI) children (GMFCS-E&R) [2,3]. Respiratory complications are common and aspiration pneumonia is a leading cause of death in this group [4]. Pharmacological management of GERD is often ineffective in children with severe neuro-disability, and anti-reflux surgery may be required [4]. Laparoscopic fundoplication remains the most frequently performed operation, with a consistent rate of recurrence. “Re-do” surgery is required in a range of patients between 6% and 14% at 20–36 months of follow-up. Other post-operative problems include high rates of dysphagia, gagging and retching, as well as dumping syndrome and gas bloating [4,5]. Rates of failure of redo-Fundoplication is variable in different series from 21% [5] to 42% [6] depending on definitions used, type of patients considered and duration of follow-up.

Because of the marked frequency of recurrent reflux following fundoplication, Bianchi (1997) suggested total esophagogastric dissociation (TEGD) as a definitive primary procedure for NI patients [7]. Experience of primary TEGD from specialized centers confirmed the complete resolution of symptoms, with significant post-operative improvement in clinical conditions and weight gain [8,9].

These results stimulated reflections around other challenging conditions that significantly increase the risk of GERD: esophageal atresia (EA), esophageal strictures, tracheoesophageal cleft (TEC), caustic burn injuries (CBI), microgastria and congenital diaphragmatic hernia (CHD). It is well-known that chronic gastrocolic reflux appeared in 35–70.8% of esophageal replacements with colonic interposition (esophagocoloplasty), especially in cases of anterior cologastric anastomosis [10,11].

In 2006, Bianchi suggested TEGD for neurologically normal children to resolve specific and difficult situations such as microgastria, cologastric reflux, and loss of the stomach from gas bloat [12]. In a variation but otherwise similar concept, a vascularized isoperistaltic (anti-reflux) 30 cm jejunal loop was interposed between the esophagus and the stomachto adequately drain or replace a dysmotile and often short esophagus consequent on esophageal atresia and its management, or a scarred esophagus from caustic or reflux injury [12]. This approach has since been adopted by other centers with similar results.

This paper seeks to present an analysis of the collected experience from the literature, on the impact of TEGD in neurologically normal children with challenging anatomical situations in order to identify selected conditions where primary TEGD could be evaluated.

## 2. Materials and Methods

A comprehensive literature review was conducted in accordance with the 2020 Preferred Reported Items for Systematic reviews and Meta-Analysis (PRISMA) [13]. A PubMed and Google Scholar electronic search without restrictions was performed using the keywords “total oesophagogastric dissociation”, “total esophagogastric dissociation”, “total oesophagogastric disconnection”, “total gastric dissociation”, “TOGD” and “TEGD”. Our literature search and review were conducted in January 2022 and included case reports, case series, original articles, and reviews written in English and published from 1997 to date. Three different reviewers independently extracted data, with discrepancies resolved by consensus. The NI children were excluded and the study focused on neurologically normal children who had either a Primary-TEGD or a Rescue-TEGD for GERD. Cases described in more than one article were considered only once.

Data were recorded in a Microsoft Excel dataset in chronological order, including study characteristics (author and publication year) and patient characteristics (main disease, surgical treatment, follow-up time and outcomes). Primary outcomes were surgical-procedure-related mortality, nutritional status and post-operative pulmonary exacerbations. Complications were classified according to the Clavien–Dindo classification and as ‘early’ or ‘late’ if they appeared within or at least 30 days after surgery. Numerical variables were expressed as mean and percentage. Study characteristics and results are summarized in tables.

## 3. Results

This literature search yielded 27 eligible full-text articles, of which 20 were excluded because 18 described NI patients and 2 contained descriptions of previously published cases. Of the selected seven articles, published between 2005 and 2021, three were case reports and four were series (Figure 1).

The study reviewed 50 neurologically normal children, between 15 days and 4 years of age, who had undergone esophagogastric or cologastric dissociation for GERD. The main diagnosis and patients’ characteristics are summarized in Table 1.

Two patients cited by Gottrand (one EA and one laryngeal diastema) were excluded from the analysis because their clinical characteristics were not described in the original article; therefore, 48 patients were considered for the purpose of the study. 

A Primary-TEGD was performed in 27 patients (56.25%). Of this, 15 presented with EA: 9 underwent esophagocoloplasty and cologastric dissociation during the same surgery, while 6 had a TEGD after esophagocoloplasty for cologastric reflux. Moreover, two out of three children with tracheoesophageal cleft type III had a Primary-TEGD at the same session as the first operation for cleft closure, and the third child 3 years later during redo surgery for partial cleft breakdown. Of three children with esophageal stricture from caustic burns, two had a Primary-TEGD at the same time as esophagocoloplasty and cologastric dissociation and a third during a redo esophagocoloplasty following necrosis of the first colonic graft. Another three patients presented severe microgastria associated with: EA (1), congenital diaphragmatic hernia with congenital short esophagus (1) and congenital esophageal stenosis (1).

Rescue-TEGD was undertaken for 21 patients (43.75%) because of failed fundoplications. Sixteen had EA: twelve underwent end-to-end anastomosis and four esophagocoloplasty. Three cases presented tracheoesophageal cleft type III; one patient was affected by CHARGE Syndrome and one presented microgastria with persistent reflux after a previous partial anterior fundoplication. An average of 2.3 anti-reflux procedures (1–5) had been performed on patients of this group before TEGD.

Forty-seven TEGDs (97.9%) were performed at open laparotomy, as described by Bianchi, and one was conducted laparoscopically with the same retro-colonic retro-gastric passage of the Jejunal Roux-en-Y loop. All (97.9%) had a gastrostomy except for one case with microgastria and normal swallowing. A Heineke–Mikulicz pyloroplasty was added on one occasion because of a vagal nerve injury at previous surgeries [18].

There was no operative mortality and the mean follow-up time was 5.5 years (range 2–10.9 years). Complications were divided into early and late and are summarized in Table 2.

According to Clavien–Dindo classification, one grade I and no grade II complications were reported in papers screened. Two children (4.1%) had early complications (Grade IIIb) and required reoperation for peritonitis and evisceration 4 and 10 days after TEGD [3].

Thirteen patients (27.1%) presented late complications (Grade IIIb): 10.4% required surgical intervention, while 16.6% required endoscopic treatments. Six months after Lap-TEGD, para-esophageal herniation of the jejunal loop through the hiatal opening was treated laparoscopically [15]. Two children underwent laparotomy for bowel obstruction and one child needed surgery at 3 and again at 3.5 years after TEGD for intestinal evisceration and herniation of the jejunal Roux-en-Y loop, respectively. Six months after TEGD, another child with EA and native esophagus underwent resection and redo-anastomosis after three pneumatic dilatation failures for stenosis of the esophagojejunal anastomosis.

Five patients with esophagojejunal anastomotic stenosis and three patients with esophagocolic anastomotic stenosis responded to repeated endoscopic balloon dilatations and Savary-Gillard bougienage. Two of them also had Mitomycin C applied to the stricture. Barrett’s esophagus (gastric metaplasia without intestinal metaplasia or dysplasia) was observed in three children 8 and 9 years after TEGD, but it was unclear whether this had been present at the time of TEGD [17].

TEGD related mortality was 2.08%, with one child (primary TEGD in congenital diaphragmatic hernia with short esophagus) dying 2 years after surgery from small bowel herniation in the thorax (Grade V) [13]. Nonsurgical mortality reached 16.6% (eight patients) from acute respiratory failure at 3 to 18 months after TEGD. Ten patients had severe GERD-related chronic respiratory disease at the time of TEGD, and two of them needed a tracheostomy a few months after surgery. One of these patients, born with a tracheoesophageal cleft, had a primary TEGD at 15 days of life. She had a tracheotomy at 3 months and died from septic shock after four operations for recurrent tracheoesophageal fistula. A second child with EA type III underwent a rescue TEGD after Nissen fundoplication and anastomotic stenosis for severe respiratory disease at 2.4 years, with a tracheotomy performed at 2.5 years, but she died soon after of respiratory failure [3]. A third patient (severe microgastria) had a tracheostomy placed at 1 month of age for an associate diagnosis of Pierre Robin sequence. She underwent primary TEGD at 11 months and approximately 1 year post-operatively, her tracheostomy was removed [16].

At follow-up, all the survivors had an improved general condition, with resolution of GERD and weight gain. Respiratory symptoms almost completely disappeared in 26 patients (54.17%), while 9 (18.7%) had persistent mild respiratory symptoms with a reduced incidence of chest infections. 

Twenty-four children (50%) were weaned off enteral nutrition and fed entirely orally “ad libitum”, with seven (14.6%) having their gastrostomy removed in a range of follow-up between 3 and 5.6 years. Twenty (41.67%) were able to feed orally but accepted only a reduced intake and needed enteral supplements, while one child remained on total enteral nutrition. In eight children, iron deficiencies were found, and five experienced malabsorption of Vitamin B12 and fat-soluble vitamins and required oral supplementation. All of the children with digestive malabsorption were weaned from enteral nutrition [3].

## 4. Discussion

Several congenital conditions such as esophageal atresia, congenital diaphragmatic hernia and tracheoesophageal cleft are associated with an increased incidence of GERD and esophageal stricture, especially in difficult primary repairs. The reason is probably related to the persistence of esophageal anatomic anomalies and/or physiologic disturbance that may even be aggravated by previous surgeries [11]. This mechanism is clearly evident in patients with esophageal caustic burns or long gap esophageal atresia when esophageal replacement is needed. The colon has been the organ most commonly used; however, most series have been fraught with major complications and conduit loss. Following esophagocoloplasty, reflux into the colonic neo-esophagus is the norm and may lead to dysphagia and delayed passage and stagnation of oral feeds in the graft [17]. Nowadays, different authors switched to gastric or jejunal transpositions with a slightly lower mortality and a substantial reduction in cervical anastomotic leakage [19].

Nissen fundoplication was often proposed as a first anti-reflux surgery. The high failure rate of this procedure is linked to wrap herniation through the esophageal hiatus in the diaphragm and to wrap disruption because of tension in the shortened esophagus and the increased resistance caused by the wrap itself.

In 1997, Bianchi proposed TEGD as an alternative, preferably first-line management, for gastroesophageal reflux in NI children [6]. The esophagus is transected from the stomach, which is closed off and is only accessible by gastrostomy. Bowel continuity is established by esophageal anastomosis to a transposed isoperistaltic 30 cm jejunal loop that is also anti-reflux and eliminates food aspiration. In 2006, TEGD, as a primary or a rescue procedure, was proposed for neurologically normal children with severe GERD caused by complex cases of long-gap esophageal atresia, tracheoesophageal cleft type III, congenital diaphragmatic hernia with short esophagus, congenital esophageal stenosis and microgastria [12], and other reports followed.

The majority of patients considered underwent several surgical procedures and failed fundoplication attempts before TEGD, with simultaneous worsening of respiratory function that led to a significantly higher risk of death during TEGD. For this reason, some authors suggest that TEGD should be performed not on a rescue basis but earlier, as a preventive measure [19]. It is, however, clear that is a complex procedure with carefully selected indications. Tracheoesophageal cleft type III is an example of a surgical challenge in which primary TEGD could be considered and performed at the time of cleft closure [18].

However, in patients with available esophagus or transposed colon and good anatomical conditions, it is preferable to prevent the “dissociated state” and to consider the interposition of a 30 cm jejunal loop to act as an anti-reflux mechanism between the esophagus or colon and the stomach, retaining normal bowel continuity [12]. This opportunity is an alternative solution in these highly challenging situations that will make TEGD more attractive and easily accepted by parent and by the child.

In all but one case considered in the study, the surgical approach was by open laparotomy. The one child who had a laparoscopic procedure required reoperation at 6 months for para-esophageal herniation of the loop through the hiatal opening [15]. Pyloroplasty was only added if there was concern regarding vagal nerve integrity or if the pre-operative upper gastrointestinal study showed delayed stomach emptying.

Oral nutrition has been well tolerated in patients with normal swallowing. Children who underwent TEGD are fed as children with total gastrectomy with smaller but more frequent feedings. Adaptation is easier when the surgery is performed at a younger age [14].

In seven cases, gastrostomy was removed even if this choice also removed the option of long-term direct gastric monitoring [14].

Enteral weaning failure has been related to limited oral intake with insufficient growth or malabsorption when food was given orally.

Nutritional follow-up is particularly important both for malnutrition and to prevent the more common excessive weight gain in patients fed through gastrostomy. Patients who are no longer on enteral nutrition may develop complications comparable to those following a Roux-en-Y gastric bypass developing deficiencies of iron, vitamin B12, fat-soluble vitamins (A, D, E), folate, calcium, and thiamine deficiency (absorbed in the proximal duodenum) that may lead to irreversible neurological damage, such as Wernicke encephalopathy or peripheral neuropathy. Supplementation with iron, folate, vitamins and micronutrients should be proposed if necessary [3].

Long-term endoscopic follow-up into adulthood should be carefully considered in TEGD patients and cannot be based on symptomatology alone. Gottrand et al. reported four patients, originally with esophageal atresia, who developed the following complications 8–9 years after TEGD: esophagojejunal anastomotic stricture (1), gastric metaplasia (1) and combined esophagojejunal stricture and gastric metaplasia (2). Other descriptions of esophagojejunal anastomotic stenosis were reported by De Lagausie and Madre, but no other descriptions of gastric metaplasia as a long-term complication of TEGD are present. However, these three patients had severe GERD before TEGD and metaplasia may already have been present at the time of surgery. The reasons for anastomotic strictures are unclear; bile esophagojejunal reflux could play a role and a short Roux-en-Y loop could explain these findings [17].

Following multiple surgeries, TEGD can prove technically challenging, and for this reason its use has been limited to specialists in tertiary centers. In 2020, we published a less-complex technical modification (M-EGD) of the original TEGD with the aim of offering a technically easier and reproducible method that avoids the major difficulties of para-esophageal surgery and a high esophagojejunal anastomosis. The stomach is detached from the esophagogastric junction with an articulated 5 mm stapler, leaving a 5 mm strip of stomach attached to the esophagus. The proximal jejunum was transected with a stapler and an isoperistaltic jejunal Roux passed through a window in the transverse mesocolon and behind the stomach to reach the esophagus at the lesser curve. An enterotomy is performed 1 cm above stapled gastric cuff, and a mechanical end-to-side anastomosis was created between this and the jejunal loop. A jejuno-jejunal anastomosis restores bowel continuity. We support that the retained 5 mm gastric cuff at the end of the esophagus allows a more secure anastomosis and reduces the risk of post-operative breakdown and leakage [20].

Our results confirmed that TEGD is a major surgery capable of short- and long-term complications, but with an acceptable mortality (2.08%). Therefore, it has been confirmed as the only definitive solution to the challenging situations brought about by GERD.

For this reason, primary TEGD should be carefully evaluated, since delaying surgery demonstrates a worsening of respiratory function with a greater risk of consequent mortality.

All of the surviving patients achieved good enteral autonomy with a good percentage of patients on oral nutrition and adequate growth. A close gastroenterological follow-up is needed to understand long-term outcomes where these patients have a normal life expectancy.

This review presents some biases due to missing data in several articles screened. Moreover, complications and outcomes are often not associated with a patient’s main diagnosis but considered cumulatively in papers. Similarly, we report the age of patients considered as mean and range because the lack of data for a singular patient makes it impossible to recalculate median and SD. Meta-analysis was not performed because the studies considered are too heterogeneous; for this reason, results were analyzed in a descriptive way only.

## 5. Conclusions

Total esophagogastric dissociation or cologastric dissociation are surgical procedures that still require analysis at long-term follow-up. Present evidence suggests that they are effective at controlling or eliminating the significant reflux that complicates major congenital abnormalities and debilitating acquired conditions such as caustic esophageal injury. This review stimulates discussion around the introduction of primary-TEGD in neurologically normal children in order to avoid pulmonary failure, where serious reflux that can damage pulmonary function is unlikely to be reliably controlled by conventional methods. Rescue-TEGD offers a solution when other techniques have failed. We strictly believe that these complex patients should be centralized in a limited number of expertise centers.

## Figures and Tables

**Figure 1 children-09-00999-f001:**
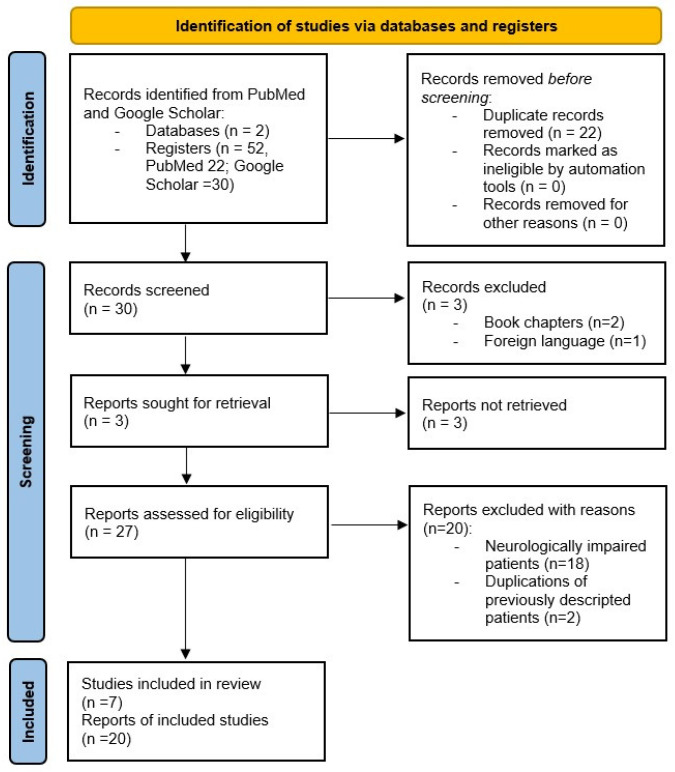
PRISMA Flow Diagram [13].

**Table 1 children-09-00999-t001:** Patients’ description.

Author	*n*°	Sex		Mean Age Age Range	Primary Diagnosis	Previous Anti-Reflux Surgery			Gastrostomy (*n*°)	Gastrostomies Removed (*n*°)
		M	F			YES (*n*°)	NO (*n*°)	Redo NF (*n*°)		
De Lagausie [14]	13	6	7	35 m (14 d–218 m)	9 EA (6 ECP)	NF (2)	7	Y (2)	Yes (13)	Yes (2)
					2 TEC	0	2	N		
					2 CBI (2 ECP)	0	2	N		
Lall [12]	11	7	4	52.7 m (24 m–72 m)	3 EA (ECP)	0	3	N	Yes (11)	No (7) but oral feeding “ad libitum”
					2 EA short gap	NF (2)	0	N		
					1 Gastrectomy	0	1	N		
					Microgastria	0	1	N		
					Congenital stenosis	0	1	N		
					Short esophagus and CDH	0	1	N		
					2 TEC	NF (2)	0	N		
Boubnova [15]	1	0	1	54 m	EA	NF(1)	0	Y	Yes (1)	No (1)
Madre [3]	17	11	6	14.5 m (15 d–206.5 m)	7 ECP	NF (4)	3	N	Yes (17)	Yes (4)
					5 EA	NF (4)	1	N		
					2 TEC	NF (1)	1	Y		
					Microgastria	0	1	N		
					CBI	0	1	N		
					CHARGE	NF (1)	N	Y		
Kunisaki [16]	1	0	1	11 m	Microgastria	0	1	N	Yes (1)	Yes (1)
Gottrand [17]	4	4	0	12.9 m (1.5 m–14 m)	2 EA type III	NF (1)	1	N	Yes (4)	No (4)
					1 EA and Microgastria	NF (1)	0	N		
					1 EA Type I	NF (1)	0	N		
	2 *	N/A	N/A	N/A	1 EA	N/A	N/A	N/A	N/A	
					1 Laringeal diastema	N/A	N/A	N/A	N/A	
Hattori [18]	1	1	0	7 m	Microgastria	PANF (1)	0	N	No (1)	

*n*° (number of patients), M (male), F (female), EA (esophageal atresia), ECP (esophagocoloplasty), TEC (Tracheoesophageal cleft), CBI (caustic burn injuries), CDH (Congenital diaphragmatic hernia), NF (Nissen fundoplication,), PANF (partial anterior fundoplication), N/A (not available), d (day), m (months), * (not analyzed).

**Table 2 children-09-00999-t002:** Post-operative complications.

Author	N° Patients	Early Complications (Main Diagnosis; Time after TEGD)	Early Complications’ Treatment	Late Complications (Main Diagnosis; Time after TEGD)	Late Complications’ Treatment
De Lagausie [14]	13	0	0	1 Esophagocolic anastomotic stenosis (N/A; N/A)	Pneumatic dilatations
				1 Esophagojejunal anastomotic stenosis (N/A; m 3)	Pneumatic dilatations
Lall [12]	11	0	0	1 Peritonitis after small bowel herniation in the thorax (CHD with short esophagus; y 2)	Patient’s death
Boubnova [15]	1	0	0	1 Small bowel paraesophageal herniation (EA; m 6)	Laparoscopic correction
Madre [3]	17	1 Peritonitis (Microgastry, day 4)	Laparotomy	Bowel obstruction (EA, m 2)	Laparotomy
				2 Esophagojejunal anastomotic stenosis (2 EA, m 6)	1 Pneumatic dilatation
					1 Redo anastomosis
		1 Evisceration (EA, day 10)	Laparotomy	2 Esophagocolic anastomotic stenosis (1 EA y 1,4.5; 1 CBI y 1, 2.5)	Pneumatic dilatations with Mitomycin C application
				1 Eventration and herniation of jejunal loop (EA; y 3; y 3.5)	Laparotomy
Kunisaki [16]	1	0	0	1 Adhesive small bowel obstruction (Severe microgastria; N/A)	Enterolisis
Gottrand [17]	6	0	0	1 Esophagojejunal stenosis (EA type I; y 1)	Endoscopic dilatation with Mitomycin C application
				1 Barret esophagus (EA type III; y 10)	0
				2 Combined esophagojejunal stenosis and gastric metaplasia (2 EA type III; m 12; y 9)	Endoscopic dilatation, PPI and follow-up
Hattori [18]	1	0	0	0	0

‘Early’ or ‘late’ if they appeared within or after 30 days post-operatively; CHD (Congenital diaphragmatic hernia); EA (esophageal atresia); m (months); y (years); N/A (not available).

## Data Availability

The data that support the findings of this study are available on request from the corresponding author (E.N.).

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
