# Peer review of "Total Esophagogastric and Cologastric Dissociation in Neurologically Normal Children: Systematic Review"

_children, 2022, doi:10.3390/children9070999_

Round 1
Reviewer 1 Report
Peer Review Children 2022
The present study reports as a systematic review experiences with total esophagogastric and cologastric dissociation in neurologically normal children.
Major points: In this very rare indication for an extensive procedure in neurologically normal children asystematic literature research was done. The selected forty-eight patients were described and the outcome discussed.
Abstract: Results and Results section: is well written and shows in a short and good summarized manner the aims and results of the study.
Methods: A comprehensive literature review was conducted following the PRISMA guidelines. The authors are mentioning by themselves that this review presents some publication biases due to missing data in several articles screened.
Results section: The main message is that at follow up all the survivors had improved general condition. The mortality rate was 2.08% with one child dying 2 years after surgery.
Discussion: The discussion is very worthful including all the necessary criteria for assuming in the end the value and risk of this rare indicated operation. It is helpful that in the discussion technical adaptions of the procedure are mentioned.
Author Response
Thank you for your valuable comments that fully capture the spirit of our work.
Reviewer 2 Report
This is a systematic review of children who underwent total esophagogastric dissociation. Since the procedure has been considered for neurologically impaired children suffering from GERD, it is worth publishing to show that it would be an alternative for GERD in children with challenging conditions other than neurologic impairment.
Please consider below in the revised manuscript.
Major
1, I wonder if all the patients included in this study were neurologically normal children. The patients with challenging conditions may have had neurological problems as well. If the patients were not clearly described as neurologically normal children in those studies, it would be better not to use the term neurologically "normal."
2, According to table 1, a primary -TEGD (no previous anti-reflux surgery) was performed on 26 patients, and a rescue -TEGD (previous anti-reflux surgery) was performed on 25 patients, which was different from the statements in the result.
3, Data should be shown as median and range or mean and SD, not mean and range.
2, In the Discussion, the authors should describe the results and interpretations compared with previous studies. Although this study was descriptive, the authors could have summarized what they found in this study to highlight their insights. However, they described the background for TEGD in the first four paragraphs. The readers would not understand what the important results of this study were.
Minor
Line 87-89 should be in the limitation.
In Table 1,
Boubnova reported one female patient. It should be M0, F1.
Was age shown by months? If so, please indicate it.
Only one patient underwent Pyeloplasty, so it is unnecessary on the table.
All but one patient had gastrostomy. It would better show the number of patients who weaned off gastrostomy after TEGD.
